# Quantitative Spatiotemporal Mapping of Lipid and Protein Oxidation in Mayonnaise

**DOI:** 10.3390/antiox9121278

**Published:** 2020-12-15

**Authors:** Suyeon Yang, Aletta A. Verhoeff, Donny W. H. Merkx, John P. M. van Duynhoven, Johannes Hohlbein

**Affiliations:** 1Laboratory of Biophysics, Wageningen University & Research, Stippeneng 4, 6708 WE Wageningen, The Netherlands; suyeon.yang@wur.nl (S.Y.); donny.merkx@unilever.com (D.W.H.M.); 2Unilever Global Foods Innovation Centre, Plantage 14, 6708 WJ Wageningen, The Netherlands; lia.verhoeff@unilever.com; 3Laboratory of Food Chemistry, Wageningen University & Research, Bornse Weilanden 9, 6708 WG Wageningen, The Netherlands; 4Microspectroscopy Research Facility, Wageningen University & Research, Stippeneng 4, 6708 WE Wageningen, The Netherlands

**Keywords:** lipoprotein granules, co-oxidation, α-tocopherol, l-ascorbic acid, antioxidant, pro-oxidant, segmentation, tracking, oxidation rate

## Abstract

Lipid oxidation in food emulsions is mediated by emulsifiers in the water phase and at the oil–water interface. To unravel the physico-chemical mechanisms and to obtain local lipid and protein oxidation rates, we used confocal laser scanning microscopy (CLSM), thereby monitoring changes in both the fluorescence emission of a lipophilic dye BODIPY 665/676 and protein auto-fluorescence. Our data show that the removal of lipid-soluble antioxidants from mayonnaises promotes lipid oxidation within oil droplets as well as protein oxidation at the oil–water interface. Furthermore, we demonstrate that ascorbic acid acts as either a lipid antioxidant or pro-oxidant depending on the presence of lipid-soluble antioxidants. The effects of antioxidant formulation on local lipid and protein oxidation rates were all statistically significant (*p* < 0.0001). The observed protein oxidation at the oil–water interface was spatially heterogeneous, which is in line with the heterogeneous distribution of lipoprotein granules from the egg yolk used for emulsification. The impact of the droplet size on local lipid and protein oxidation rates was significant (*p* < 0.0001) but minor compared to the effects of ascorbic acid addition and lipid-soluble antioxidant depletion. The presented results demonstrate that CLSM can be applied for unraveling the roles of colloidal structure and transport in mediating lipid oxidation in complex food emulsions.

## 1. Introduction

Unsaturated fatty acids are nutritionally important [1] and often consumed in the emulsified form, such as infant formula, milk, mayonnaise, and dressings. The physical stability of these food emulsions is enhanced by surface-active lipids and proteins at the oil–water interface [2,3]. The presence of these emulsifiers, however, compromises chemical stability by promoting lipid oxidation, which leads to off-flavors and compromised nutritional value [4]. Consequently, understanding and controlling lipid oxidation is an area of active research, with studies predominantly focusing on the chemical fate of lipids upon oxidation [2,5,6,7]. At the initial stage of lipid oxidation, unsaturated lipids produce lipid free radicals. During the propagation phase, these radicals react with oxygen and produce peroxyl radicals, which produce hydroperoxides that ultimately degrade to volatile off-flavors. In food emulsions, it is known that these reactions are catalyzed at the droplet interface and that antioxidants can be active in the lipid droplet phase, the water phase, and at the oil–water interface [8,9]. Furthermore, there is growing evidence that colloidal surface heterogeneity and the transport of oxidation reaction intermediates also play a critical role in the onset and the rate of oxidation [10,11].

Several routes for the role of mass transport of lipid oxidation in food emulsions have been proposed in the literature [11]. In comparison to the diffusional transport of individual hydrophobic lipid oxidation intermediates in the aqueous phase, droplet–droplet contacts could present a more efficient exchange mechanism. An even more efficient way to transfer oxidation intermediates between droplets could be micellar transport, aided by the presence of emulsifiers that are not absorbed at droplet interfaces. This micelle-assisted transfer has been proposed as the main accelerator of lipid oxidation, as the amount of emulsifiers used for stabilizing food emulsions is sufficient to form micelles [2]. According to this micelle-assisted transfer hypothesis, the rate of lipid oxidation can be modulated by controlling the concentration of emulsifiers.

Besides propagation of lipid oxidation, free lipid radicals can also engage in the oxidation of proteins present at droplet interfaces and in the continuous water phase. There is, however, scarce information on the cause–effect relationships in lipid–protein co-oxidation [12]. Progress has been impeded by a lack of methods that can provide spatial insights in the co-oxidation of lipids and proteins.

Whereas the effects of various ingredients that delay lipid oxidation in food emulsions have been widely studied at the ensemble level [6,8,13], there is scarce or even ambiguous support for current hypotheses on the spatial and temporal heterogeneity of lipid oxidation [14]. Variation in oil droplet size has been proposed as a factor in the spatiotemporal heterogeneity of lipid oxidation. Some publications suggested that smaller lipid droplets have a higher rate of oxidation due to their larger surface area to volume ratio [5,15,16] whereas others showed that the droplet size itself has no significant effect on the rate of lipid oxidation [17,18,19,20,21]. In the water phase, the composition of anti/pro-oxidants, including ascorbic acid, unadsorbed proteins, and metal iron, can inhibit or promote the formation and diffusion of reactive species dependent on their concentration and the pH value of the emulsions [13,22]. These anti/pro-oxidants further influence the characteristics of the interfacial layer and can change the oxidation rate by having the role of metal chelators as well as free radical scavengers at the oil–water interface [2].

The fluorophore BODIPY 665/676 and other dyes of the series have been widely used for studying lipophilic structures as it is sensitive to peroxyl/alkoxyl radicals [23,24,25,26]. Recently, the role of mass transport in lipid oxidation between emulsion droplets was studied using flow cytometry [24]. This technique, however, does not allow the individual droplets to be followed in situ over time nor does it distinguish whether and to which extent oxidation occurs at the interface or the continuous phase. 

Here, we present a method that can assess oxidation in a time-dependent manner in three distinct phases: (1) lipid oxidation in the dispersed oil phase, (2) protein oxidation in the continuous water phase, and (3) protein oxidation at the oil–water interface. We assessed lipid oxidation of oil droplets in mayonnaise using BODIPY 665/676, which changes its fluorescence emission from red to green/yellow after interaction with radicals, and we evaluated protein oxidation by detecting protein auto-fluorescence [27,28,29] in both the aqueous phase and interface. 

## 2. Materials and Methods

### 2.1. Materials

The lipophilic and oxidation-sensitive dye BODIPY 665/676 was purchased from Thermo Fischer (Waltham, MA, USA). l-ascorbic acid, sodium chloride (>99.5%, EMSURE^®^), and alumina power (Alumina N—Super I) were obtained from Aldrich-Europe (Darmstadt, Germany), MilliporeSigma (Burlington, MA, USA), and MP EcoChrom™ (Eschwege, Germany), respectively. Spirit white vinegar (4%), soybean oil, and egg yolk containing 8% (*w*/*w*) NaCl were purchased from a local store. Demineralized (demi) water was used for all experiments.

### 2.2. Preparation of Oil Samples

Soybean oil was stripped using alumina powder to remove lipid-soluble antioxidants [22]. The powder was mixed with the oil at a volume ratio of 1:2 in Falcon tubes and the mixture was shaken in the dark for 24 h. The mixture was then centrifuged at 2000× *g* for 20 min to separate the stripped soybean oil. The oil was collected, and the same centrifugation procedure was repeated to ensure complete removal of the alumina powder. BODIPY 665/676 was dissolved in soybean oil or stripped soybean oil to a final concentration of 1 µM, which is too low to act as an antioxidant [24].

### 2.3. Preparation of Emulsions

Mayonnaise was prepared using a Silverson mixer: 78% (*w*/*w*) of soybean oil, 5% (*w*/*w*) of egg yolk with 8% (*w*/*w*) NaCl, 0.72% (*w*/*w*) of sodium chloride, 11.75% (*w*/*w*) demi water, and 4.53% (*w*/*w*) spirit vinegar. Egg yolk, salt, demi water, and the half volume of vinegar were premixed in a 500-mL jar and mixed at 2000 rpm for 20 s. Oil containing 1µM BODIPY 665/676 was slowly added and mixed at 8500 rpm for 4 min. Spirit vinegar was added and mixed further for 2 min. For the manipulation of oxidation via a water-soluble agent, we prepared a stock solution of 2.1 g of l-ascorbic acid in 6.5 mL of demi water. To obtain a 10 mM final concentration of l-ascorbic acid, 1.1 mL of stock solution were gently stirred into 180 g of mayonnaise.

Mayonnaises were prepared with either soybean oil or stripped soybean oil, in which native oil-soluble antioxidants were removed. Additionally, we performed measurements with and without ascorbic acid (Table 1).

### 2.4. Accelerated Oxidation of Samples

For imaging measurements, the emulsions (200 µL each) were placed into separate μ-slide wells and stored at 30 °C to accelerate the oxidation (Figure 1A). For the measurements in the fluorescence spectrophotometer, 15 samples with 1.5 mL of emulsions in 2-mL Eppendorf tubes were stored at 30 °C. Every day, a sample was placed in a −20 °C freezer to stop oxidation. After 15 days, the samples were thawed, resulting in the separation of the oil and water phases. Clean oil phases were obtained by centrifuging the phase-separated emulsion at 2000× *g* for 5 min and removing the water phase.

### 2.5. Preparation of Sample Carrier

To monitor lipid droplets at the same location over time, we engraved an “x” into the bottom of a glass sample carrier (μ-slide 8-well glass bottom, ibidi^®^, Munich, Germany) using a diamond knife (Figure 1A). The carrier was then plasma cleaned (1 min) to remove organic contaminations from the glass and to prevent absorption of oil droplets to the surface. After placing the samples in the carrier, a lid was glued onto it using an epoxy resin to prevent evaporation. The carrier was then placed on a confocal laser scanning microscope (CLSM, Leica SP8, Wetzlar, Germany) such that the same position can be revisited on consecutive days.

### 2.6. Confocal Laser Scanning Microscopy (CLSM)

Monitoring of lipid and protein oxidation was carried out on a CLSM (Leica SP8, Wetzlar, Germany) equipped with a 63× NA = 1.2 water immersion objective (HC PLAPO CS2, Leica, Wetzlar, Germany) and a white-light laser with user-selectable excitation wavelengths. The scanning format was 512 × 512 pixels (123 µm by 123 µm) and the line-scanning speed was set to 100 Hz. The excitation wavelength was set to 561 nm to detect oxidized lipids with BODIPY 665/676 (detection range from 580 to 660 nm, green channel) or 640 nm to measure non-oxidized lipids (detection range from 660 to 750 nm, red channel). To detect protein oxidation, samples were excited at 488 nm and fluorescence emission was detected between 500 and 560 nm (blue channel). Images were tracked 15 days for sample I and II and 10 days for sample III and IV.

### 2.7. Spectrophotometry

The emission spectra of BODIPY 665/676 were measured with a fluorescence spectrophotometer (Fluorlog 322, Horiba, Kyoto, Japan). The laser excitation and the detection range of emission were the same as with the measurements in CLSM (λ_ex_561/640 nm, λ_em_580–660/660–750 nm). The final concentration of BODIPY in oil was calculated from the absorption coefficients determined with an absorption spectrophotometer (Cary4000, Santa Clara, CA, USA).

### 2.8. Segmentation and Tracking

For segmentation in StarDist [30], we set the percentile low and high value to 1 and 99.8, respectively. The probability/score threshold was set to 0.5 and the overlap threshold was 0.4. For the neural network prediction, we used the versatile (fluorescence nuclei) model. For tracking in Trackmate [31], the calibration settings were 1 pixel for pixel width, pixel height, and voxel depth. The time interval was set to 1 frame and a LoG detector was used. The estimated blob diameter was set to 14 pixels and the threshold was set to 0. A median filter and sub-pixel localization option were used in combination with a simple LAP tracker option. Linking max distance and gap-closing max distance were set to 15.0 pixel, and gap-closing max frame gap was set to 2. The filters were set on tracks above 9.9 (numbers of spots in track). After tracking, the label image was exported with the option of only spots in tracks. For segmentation and tracking analyses, sample IV (SSO + aa) was not included as the fast oxidation led to extensive droplet coalescence.

### 2.9. Gompertz Curve Fitting

Local oxidation rates were determined using the Gompertz function f(t)=a exp(−bexp(−ct)), with *a, b,* and *c* representing the asymptote of the curve, the displacement along the x-axis, and the growth rate. Lipid and protein oxidation data were fitted using least-squares regression with the Excel-solver (Version 2020, Microsoft Corporation, Redmond, WA, USA). The obtained growth rates were used as indications for the local oxidation rate of lipids in oil droplets and proteins at the interface, respectively.

### 2.10. Statistical Analysis

ANCOVA (analysis of covariance) was used to study the combined effect of antioxidant formulation and inverse of droplet radius. Antioxidant formulation (sample I, II, and III) was treated as categorical factors and the inverse of droplet radius as a continuous covariate. The local lipid/protein oxidation rate was the dependent variable. All statistical analyses were done with JMP software (Version 15) from SAS (Cary, NC, USA). The significance level α was set to 0.05.

## 3. Results

### 3.1. Confocal Microscopy Allows Spatiotemporal Mapping of Lipid and Protein Oxidation

We compared the oxidation kinetics for four different emulsions (SO, SO + aa, SSO, and SSO + aa, see Table 1) in a modified sample carrier (Figure 1A). The carrier allowed us to track oil droplets at the same location over many days on a confocal laser scanning microscope (CLSM). As a proxy of lipid oxidation, we used BODIPY 665/676. In total, we monitored three spectral regions (see the materials and methods) to obtain spatially and temporarily resolved information on lipid and protein oxidation at the interface, in oil droplets, and in the continuous phase. The three spectral regions were then overlaid to obtain a single pseudo-color coded CLSM image for each condition (Figure 1B). 

We will first *qualitatively* describe our observations, before performing further quantitative analysis. On day one, samples I-III (SO, SO + aa, and SSO) showed similar fluorescence emission characteristics, with most of the FL originating from the red emission channel, indicating a similar non-oxidized state. Sample IV (SSO + aa), however, showed already increased intensities in the green and blue emission channel, indicating that lipids and proteins are already highly oxidized. On the third day, there was a modest increase of green emission in samples I-III and no increase in the blue channel for samples I and III (no aa). Whereas sample II (SO + aa) showed a modest increase in the blue channel for the aqueous phase, sample IV (SSO + aa) showed high intensities in both the blue and the green detection channels. On day seven, a small increase of green emission was visible for sample I (SO) and a larger increase for sample III (SSO). Sample II (SO + aa) showed an increase in blue fluorescence at the interface of oil droplets. After 15 days at 30 °C, sample I (SO) showed dominant green fluorescence; in contrast, no significant increase in green fluorescence was seen in sample II (SO + aa). For samples III and IV (SSO, SSO + aa), the emulsion broke after 15 days, hence no data are shown. To support the general changes of lipid and protein oxidation, we further recorded data at different positions in the same sample carrier and with an independent repeat in a different carrier as well (Appendix A
Appendix A).

To comprehend the effect of stripping the soybean oil, close-up images of the areas depicting samples I and III (SO and SSO) in the absence of ascorbic acid are presented in Figure 1C. Both samples showed similar levels of red emission FL on day one. Interestingly, both green and blue emission, indicating oxidized proteins, were higher after seven days in sample III (SSO) than in sample I (SO). To further evaluate the role of ascorbic acid, samples I and II (SO and SO + aa) were compared. Lipids were not fully oxidized even after 15 days and blue spots appeared to increase heterogeneously around the oil droplets, indicating that ascorbic acid acted as a protein pro-oxidant in sample II (SO + aa, Figure 1D). The finding of ascorbic acid acting here as a lipid antioxidant is further supported by the spectral ensemble measurements in the Appendix A.

### 3.2. Quantitative Spatiotemporal Analysis of Lipid and Protein Oxidation Maps

The analysis of intensity changes in individual droplets as a function of time is not a trivial task. Challenges involve the identification and separation of individual droplets from the water phase in emulsions and the tracking of individual droplets over many days. To solve this, we first segmented the droplets from CLSM data using the deep learning-based software 2D StarDist [30]. To this end, we used the red channel of the CLSM images for segmentation (Figure 2A,B) and applied the segmentation mask to the original three-color images using MATLAB R2019a. This approach allowed us to obtain three-segmented channels representing (Figure 2C) (1) non-oxidized lipids in oil droplets (red channel, ex 640 nm), (2) oxidized lipids in oil droplets (green channel, ex 561nm), and (3) oxidized proteins at the interface (blue channel, ex 488 nm). Figure 2C further shows the image containing the oxidized proteins in the aqueous phase obtained by applying and inverting the segmentation mask. After segmentation, the number of droplets was 600, 601, and 910, respectively, and the average radius size of droplets was 2.2, 2.1, and 1.8 µm for sample I-III (Table 2). 

We then tracked individual droplets using Trackmate [31]. To this end, we first stacked the segmented images representing the same location in the sample monitored over 10 days (Figure 2D). Only droplets that could be successfully tracked for the entire duration were analyzed further. As Trackmate does not support monitoring different changes in the size of the tracked objects, we assigned a tracking number to each droplet and re-processed the data if necessary (see also Appendix A
Appendix A). The number of droplets and their average radii are given in Table 2. Using the segmented and tracked images, we then analyzed the local oxidation rate of each droplet.

For oxidized proteins at the oil–water interface, sample I (SO) showed only small changes in the fluorescence intensities (mean intensity = 2.0 ± 0.7 photons per pixel) for 10 days (Figure 3A). In sample II (SO + aa), the mean intensities increased from 2.5 to 5.8 photons per pixel, showing a faster increase at the initial stage of oxidation. Sample III (SSO) did not show any changes at the initial stage (1.8 and 1.9 photons per pixel for day one and day four, respectively) but showed an increase in fluorescence after six days, leading to 7.5 photons per pixel on day 10. For oxidized proteins in the aqueous phase, the intensity changes were similar to the protein oxidation at the interface. We did not observe an increase in sample I. Sample II (SO + aa) showed the increase only in the initial stage of oxidation and at the latest stage of oxidation, sample III (SSO) had a similar oxidation level with sample II (SO + aa). 

To compare the degree of lipid oxidation in individual droplets, we divided the average BODIPY 665/667 fluorescence intensity per pixel of the green channel by the sum intensities of the green and the red channel (for the intensities in the individual channels, see Appendix A
Appendix A). The rate of lipid oxidation in sample III (SSO) is faster than the rates in samples I and II (SO, SO + aa, Figure 3B). The ratio reporting on the degree of oxidation after 10 days is 0.85 for sample III (SSO) and 0.39 and 0.32 for sample I (SO) and sample II (SO + aa), respectively. The average lipid oxidation ratio of sample I (SO) is 18% higher than the ratio of sample II (SO + aa), indicating that ascorbic acid acted as an antioxidant for lipids.

The individual oxidation rates of lipids and proteins were further quantified by fitting the data from Figure 3A and Appendix A with a Gompertz function (see the materials and methods and Appendix A
Appendix A). Here, we defined the growth parameter *c* as the local oxidation rate and calculated the rates c_lip_ and *c*_prot_ for lipids in oil droplets and proteins at the droplet interface, respectively. These local oxidation rates were plotted against the inverse of the droplet radius (Figure 3C, dots) because of the expected inverse relation between the surface area of droplets exposed to the continuous phase and the droplet radius [6,32]. For quantitative evaluation of the antioxidant effect on local lipid and protein oxidation, we performed ANCOVA (analysis of covariance) with antioxidant formulation as categorical variables and inverse droplet radius as a covariate. In all cases, the antioxidant formulation (samples I, II, and III) had a highly significant effect (*p*-value < 0.0001, see Appendix A
Appendix A). We also performed regression analyses on the lipid (*c*_lip_)/ protein (*c*_prot_) oxidation rate with the inverse droplet radius within the groups (Figure 3C, lines). For all antioxidant formulations, they showed a significant dependency of the local protein oxidation rates on the inverse of the droplet radius in all samples (*p* < 0.0001). For the lipid oxidation rates, only sample II (SO + aa) showed a significant inverse droplet size dependency (*p* < 0.0001). We note that for all samples with significant effects of the inverse droplet size on oxidation rates, the slopes were small compared to the antioxidant effects. To assess whether the weak effect of the droplet size on the local oxidation rate was due to inter-droplet exchange of dye, we mixed emulsions prepared with and without BODIPY 665/676 (Appendix A). Within 10 days, the timeframe of our oxidation experiments, the peak fluorescence intensity in undyed droplets had increased by 45%. The calculated oxidation rates, however, are determined on a shorter time frame and exchange of BODIPY665/676 between droplets may therefore only partially contribute to averaging out the effect of the droplet size on the lipid oxidation rate.

## 4. Discussion

In this study, we show that ascorbic acid takes on different roles on lipids depending on the availability of lipid-soluble antioxidants, such as tocopherol. In the non-stripped sample II (SO + aa), the antioxidant role of ascorbic acid for lipids is dominant over a pro-oxidant effect via redox cycling of transition metals, likely ferric (Fe^3+^) to ferrous (Fe^2+^) ions, at the lipid droplet interface (Figure 4A). We note that all mayonnaises in this study were prepared at low pH, where reduced binding strength to phosvitin allows for effective ferric-ferrous redox cycling [22,34]. In the stripped sample IV (SSO + aa), the ascorbic acid can no longer work in synergy with tocopherol and its pro-oxidant role becomes dominant (Figure 4B). Hence, for this emulsion (sample IV), we observed rapid lipid oxidation. This observation is in line with the pro-oxidant effect of ascorbic acid on lipids in mayonnaises prepared with fish oil, which has low concentration of α-tocopherol [22].

Whereas ascorbic acid had different roles on lipid oxidation, it always acted as a pro-oxidant on proteins in the aqueous phase and at the interface (sample II and IV). As a pro-oxidant (Figure 4B), ascorbic acid interacts with transition metals that are known to promote lipid oxidation by propagating lipid hydroperoxide (LOOH) formation at the interface of dispersed oil droplets [13,22]. Our results show that ascorbic acid also acts as a pro-oxidant for proteins at the interface and in the water phase. The heterogeneous distribution of protein oxidation at the interface in sample II (SO + aa, Figure 1D) points towards lipoprotein granules that are typically present in egg yolk of mayonnaises [35,36,37]. Additionally, here, ascorbic acid can interact with transition metals, such as iron and copper, that are known to catalyze lipid oxidation in lipoprotein particles [29]. It has been shown that lipid oxidation in lipoproteins can induce oxidation of constituent apoproteins, which becomes apparent as increased protein autofluorescence in the visible wavelength region [29]. 

We attribute the faster lipid oxidation in sample III (SSO, Figure 1B) to the absence of α-tocopherol. α-Tocopherol is naturally present in vegetable oils (sample I, SO) and is known to act as a chain-breaker in radical reactions by trapping lipid radicals and forming tocopherol radicals [6,9,38,39] (Figure 4A). Our experimental results show that in the absence of tocopherol, protein oxidation is enhanced at the interface and in the continuous phase (sample I (SO) vs. III (SSO)), with the heterogeneous distribution of oxidized proteins pointing towards the involvement of lipoprotein granules. We attribute the enhanced protein oxidation upon removal of tocopherol to the transport of lipid radicals from the oil droplets to the interface and continuous phase [12].

Several studies have suggested that the rate of lipid oxidation in emulsions depends on the droplet size typically when covering a large range of average droplet sizes [15,32]. In our case, only for sample III (SSO), a significant but weak, linear dependency of local lipid oxidation rate with the inverse of droplet size could be observed. A partial explanation can be the inter-droplet exchange of BODIPY 665/676 as indicated by the experiment where we mixed emulsions with and without dye. As a more likely explanation, we propose the inter-droplet exchange of lipid oxidation intermediates, such as hydroperoxides. Exchange of such intermediates can be facilitated by their intrinsic amphiphilic character in combination with micellization by non-absorbed emulsifiers [11]. This precludes lipid droplets to have a size dependency on the lipid oxidation rate. Moreover, we note that in our emulsions, the distribution of droplet sizes was relatively small (radius range 1–4 µm). For a larger difference in droplet radius, the aforementioned transport mechanism may be less effective in averaging out the droplet size effect for lipids. The inter-droplet exchange effect will not be less effective for proteins absorbed at the interface, which explains why we could consistently observe significant, but small, effects of the inverse droplet size on the oxidation rate of proteins at the droplet interface. 

## 5. Conclusions

This study demonstrates a CLSM imaging method that can provide spatiotemporal maps of lipid and protein oxidation in mayonnaise via changes in BODIPY 665/676 fluorescence emission and protein auto-fluorescence. Local oxidation rates of lipids within dispersed oil droplets and proteins at the interface or in the continuous water phase can be quantitively assessed via segmentation and tracking of individual droplets and fitting with a semi-empirical model. Removal of lipid-soluble antioxidants by oil stripping promotes lipid oxidation within oil droplets as well as protein oxidation at the interface. In the presence of tocopherol, ascorbic acid acts as a lipid antioxidant and as a protein pro-oxidant in the water phase. The heterogeneous distribution of oxidized proteins points towards remnant lipoprotein granules from the egg yolk used for emulsification. The spatially heterogeneous oxidation of proteins at the interface is in line with the heterogeneous distribution of micron-scale granules consisting of aggregated nanoscale lipoprotein particles. Upon depletion of lipid-soluble antioxidants, ascorbic acid acts as a pro-oxidant for both proteins and lipids. For mayonnaise protected by antioxidants, the impact of droplet size was minor compared with the effects of ascorbic acid addition and lipid-soluble antioxidant depletion. Our method can be deployed for spatially resolved assessment of lipid oxidation in heterogeneous food systems and to unravel the colloidal structure and transport mechanisms at play there.

## Figures and Tables

**Figure 1 antioxidants-09-01278-f001:**
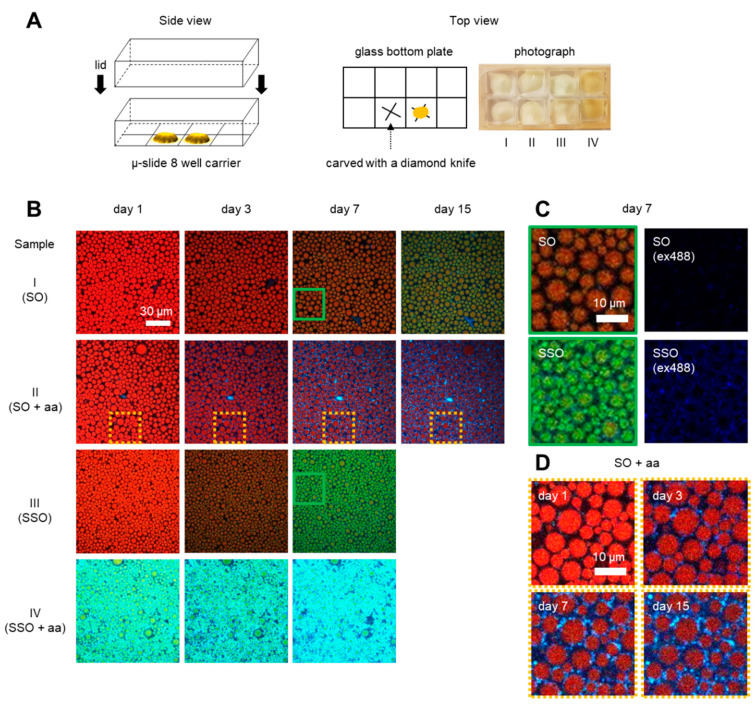
Experimental design and imaging oxidation in mayonnaise. (**A**) The side and top view of a modified sample carrier for monitoring the same field of view over time. Marks were engraved on the bottom of the glass slide using a diamond knife. The photograph shows the oxidized samples I-IV after one day at 30 °C. Sample IV shows the yellow color because of the fast oxidation. (**B**) Pseudo-color-coded CLSM images of mayonnaise containing BODIPY 665/676 stored for up to 15 days at 30 °C. Mayonnaise was made either using soybean oil (SO, Sample I), soybean oil with ascorbic acid (SO + aa, Sample II), stripped soybean oil (SSO, Sample III), or stripped soybean oil with ascorbic acid (SSO + aa, Sample IV). Images were combined from three different channels taken with λ_ex_640 nm/λ_em_660–750 nm (red), λ_ex_561 nm/λ_em_580–660 nm (green), and λ_ex_488 nm/λ_em_500–560 nm (blue). (**C**) Zoom-in of sample I (SO) and sample III (SSO) for seven days. The left column shows the combination of three channels and the right column shows only the blue channel (λ_ex_488 nm). (**D**) Zoom-in (yellow dashed box) of sample II (SO + aa) for day 1, 3, 7, and 15. Heterogeneous protein oxidation is visible throughout the continuous phase and near the oil–water interface (for details on how this figure was further processed for droplet segmentation, see Appendix A). The size of the images is 123 μm × 123 μm in (**B**) and 31 μm × 31 μm in (**C**,**D**).

**Figure 2 antioxidants-09-01278-f002:**
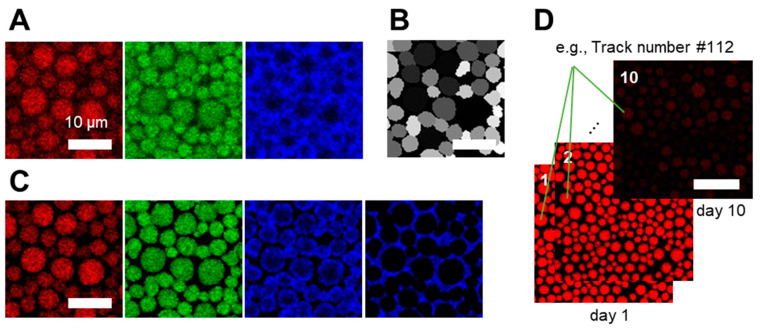
Workflow for the segmentation and tracking of individual oil droplets. (**A**) Original images (raw data) were collected in three separate channels (red, green, and blue). (**B**) Droplets were segmented using StarDist. (**C**) Segmented images were reconstructed by applying the segmentation mask (StarDist) to the original images. The blue channel images represent intensity within and outside the mask corresponding to oxidized proteins at the interface and in the continuous phase. (**D**) Segmented images that have clear borders were stacked together for tracking. Each droplet was tracked using Trackmate for up to 10 days. Tracking numbers were allocated to individual droplets. Scale bars represent 10 µm in (**A**–**D**).

**Figure 3 antioxidants-09-01278-f003:**
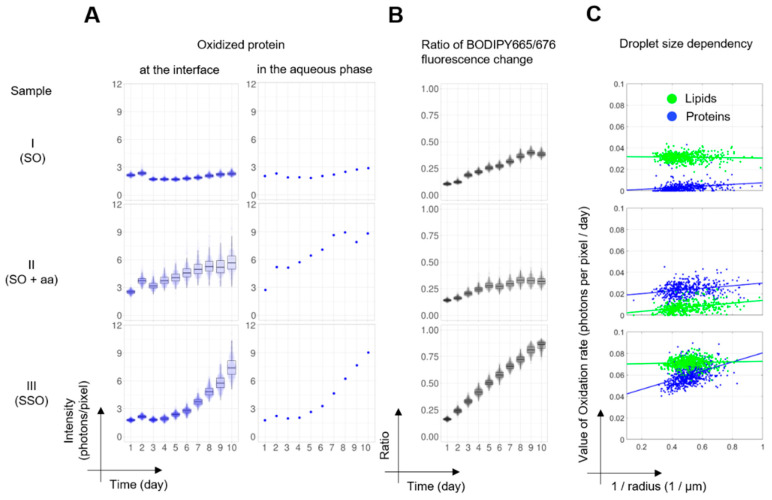
Statistical analyses of protein and lipid oxidation. (**A**,**B**) The data are visualized using PlotsOfData [33]. Each data point represents the average intensity of individual droplets. Black boxes indicate the 95% confidence interval (95CIs) of the data and the center line indicates the median value. (**A**) Intensity changes of protein oxidation at the interface and the aqueous phase. For the numbers of analyzed droplets (left column), see Table 2; for the aqueous phase (right column), we obtained a single number from the entire field of view. (**B**) The ratio of BODIPY 665/676 fluorescence intensity changes is presented as a measure of lipid oxidation. The averaged intensity per droplet from λ_ex_561 nm (oxidized lipids) was divided by the sum intensity from λ_ex_640 nm (non-oxidized lipids) and λ_ex_561 nm (oxidized lipids). (**C**) The effect of the inverse droplet size (surface to volume) on the oxidation rate of lipid within droplets (green color) and protein at/near interfaces (blue color) based on changes in fluorescence intensity. Dots show the local lipid oxidation rate as obtained from fitting the time dependency of BODIPY 665/676 color shifts and protein fluorescence changes in individual droplets with the Gompertz function. Lines show the regression analysis result of local lipid (green line) and protein (blue line) oxidation rates on the inverse of droplet sizes (see Appendix A
Appendix A).

**Figure 4 antioxidants-09-01278-f004:**
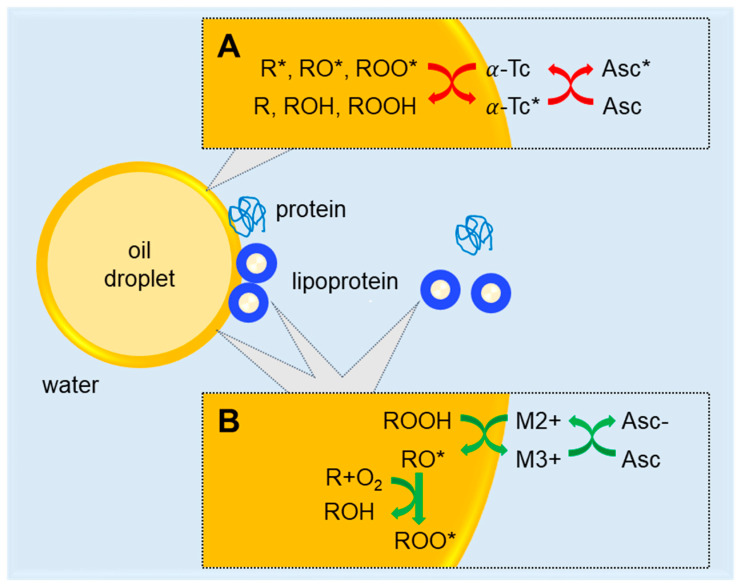
Overview of the mechanisms of lipid and protein oxidation present in oil-in-water emulsions. Oil droplets are shown in yellow; lipoprotein particles are presented with a yellow lipid core and a blue protein shell. For simplicity, these particles are not shown in their aggregated form in granules. Anti/pro-oxidant mechanisms are depicted at the interfacial droplet layer and at the lipoprotein surface. Anti- and pro-oxidant mechanisms are respectively indicated in red and green. (**A**) Ascorbic acid acts as an antioxidant for lipids by converting tocopherol radicals to tocopherol. (**B**) Pro-oxidant behavior of ascorbic acid on lipids and proteins, including lipoproteins, at the interface and the water phase. Mechanisms involve metal chelation as well as free radical scavenging. ROOH: lipid hydroperoxide; R: unsaturated lipids; R*: alkyl radical; RO*: alkoxyl radical; ROO*: peroxyl radical; ROH: hydroxy lipids; Asc: ascorbic acid; Asc−: dehydroascorbic acid; Asc*: ascorbyl radical; Tc: tocopherol; Tc*: tocopherol radical. M2+: Transition metal ions (Fe or Cu).

**Table 1 antioxidants-09-01278-t001:** The four mayonnaises studied were prepared with soybean oil (SO, Sample I), soybean oil with ascorbic acid (SO + aa, Sample II), stripped soybean oil (SSO, Sample III), stripped soybean oil with ascorbic acid (SSO + aa, Sample IV).

Sample		Lipid Soluble Antioxidant
I	Soybean oil (SO)	+
II	Soybean oil + ascorbic acid (SO + aa)	+
III	Stripped soybean oil (SSO)	−
IV	Stripped soybean oil + ascorbic acid (SSO + aa)	−

**Table 2 antioxidants-09-01278-t002:** The number of droplets and average radius size from analyses for samples I, II, and III (see Table 1).

Results	Number of Analyzed Droplets			Average Radius Size (µm) ^3^		
Sample	I	II	III	I	II	III
StarDist ^1^	600	601	910	2.2 ± 0.6	2.1 ± 0.6	1.8 ± 0.4
Re-assigned after Trackmate ^2^	484	445	641	2.2 ± 0.5	2.1 ± 0.5	2.0 ± 0.3

^1^ Minimum number of droplets per sample from segmented images that were recorded over 10 days. ^2^ Re-assigned details are explained in the Appendix A. ^3^ Standard deviation in droplet radius sizes is shown with the average radius size.

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
