# Peer review of "Quantitative Spatiotemporal Mapping of Lipid and Protein Oxidation in Mayonnaise"

_antioxidants, 2020, doi:10.3390/antiox9121278_

Round 1

Reviewer 1 Report

In this work is proposed a quantitative spatiotemporal mapping of lipid and protein oxidation in mayonnaise using CLSM imaging and fluorescence spectroscopy.

The work is well structured and understandable. I believe that it will contribute to the existing literature.

Points for correction - clarification

  1. Line 104. 8% (w/w, w/v, another?) Please write the unit.
  2. Lines 107 and 114. Not by using but using.
  3. Line 109. Not 20 mins but 20 min.
  4. Line 115. 0.72 % (w/w) of which salt?
  5. Line 118. Not 4 or 2 minutes but 4min and 2 min.
  6. Line 184. Not On day 3 but on third day.
  7. Line 199. Not 7 days but seven days
  8. Please correct the format of 207-219 lines

Author Response

Dear reviewer,

Thanks for the kind comments.

Best regards,

authors

Reviewer 2 Report

The authors assessed lipid oxidation of oil droplets in mayonnaise using the fluorophore BODIPY 665/676, which changes its fluorescence emission from red to green/yellow after interaction with radicals. The authors evaluated protein oxidation by detecting protein auto-fluorescence in both the aqueous phase and interface, and they monitored local protein and lipid oxidation by confocal laser scanning microscopy (CLSM).

Mayonnaises were prepared with soybean oil and stripped soybean oil, in which native oil soluble antioxidants were removed. Measurements were performed with and without ascorbic acid, which acts as both anti- and pro-oxidant in the water phase.

Results showed that the removal of lipid-soluble antioxidants from mayonnaises promotes lipid oxidation within oil droplets as well as protein oxidation at the oil-water interface. Ascorbic acid acts as either lipid antioxidant or pro-oxidant depending on the presence of lipid-soluble antioxidants.

The authors propose CLSM for unraveling the roles of colloidal structure and transport in mediating lipid oxidation in complex food emulsions.

The text is clear. The introduction, materials and discussions are clear and satisfying.

In my opinion, the work should be accepted in present form.

Best regards.

Author Response

(The authors gave the same response as above.)

Reviewer 3 Report

You guys have done a high-quality research concerning lipid and protein oxidation in mayonnaise using quantitative spatiotemporal mapping, i.e., CLSM method, via changes in BODIPY 665/676 fluorescence emission and protein auto-fluorescence.  The novelty of the research is high and the results also show a huge potential in future study concerning the redox reactions of perhaps any other physiologically relevant lipoproteins, e.g., HDL, LDS, and so forth.  I personally have no major objections regarding the quality of the manuscript except a few minor points that need to be addressed prior to its acceptance for publication by the journal, Antioxidants, and they are listed as follows.

It appears that α-tocopherol and ascorbic acid act simultaneously as anti-oxidants when they are both present in soybean oil.  In stripped soybean oil, SSO, where α-tocopherol is removed, ascorbic acid likely acted as pro-oxidant by reducing transition metal ions, such as Fe3+.  What is the evidence that, in SSO, transition metal ions are the dominant species that was reduced by ascorbic acid and oxidized by ROOH?  Is there any previous study, e.g., using atomic absorption spectrophotometry, AA, that explicitly showed the presence of these metal ions present in samples similar to the ones (SO & SSO) examined in this study?

In the Materials and Methods section, the statistical analyses of data should be added.

Line 77, it should be extent, instead of extend.

Line 319, it should be α-Tocopherol, instead of α-tocopherol.

Author Response

(The authors gave the same response as above.)

Reviewer 4 Report

The abstract is very interesting to read, however, is lacking some quantitative results to highlight and support the presented observations.

Regarding the keywords, I believe the number is excessive (11), and therefore they could be reduced by eliminating “lipid oxidation” and “protein oxidation”, since these terms are also present in the title and therefore they are already indexing terms for this article.

The last part of the introduction is not correct, since the introduction should end after the specification of the objective of this study. Everything else, should be in the experimental section, including table 1, which corresponds to description of what was done – experimental setup!

In table 1 – caption: The four mayonnaises studied where prepared … – the word “where” is wrong it should be “were”

Sometimes you express litters with a capital letter 8ex: mL) but other with a small letter (ex: µl) – line 123, so please correct to µL

Line 141 separate 561 nm (space between the number and the units)

The results are well presented and the supplementary materials are helpful to support the conclusions. The discussion is proper and the conclusions are supported in the findings.

The list of references is barely acceptable, although the number of old references (older than 5 years) is higher than desirable.

Refs from ³ 2015 = 13 (405)

So if the authors could improve this ratio, by citing more recent references than older ones, it would benefit the quality of the publication.

Author Response

(The authors gave the same response as above.)

Round 2

Reviewer 4 Report

After improvement attendig to the reviewer's suggestions, including my own, I consider that the manuscript presents the necessary quality to be published in this form.